# Valproic Acid Improves Antisense-Mediated Exon-Skipping Efficacy in *mdx* Mice

**DOI:** 10.3390/ijms26062583

**Published:** 2025-03-13

**Authors:** Micky Phongsavanh, Flavien Bizot, Amel Saoudi, Cecile Gastaldi, Olivier Le Coz, Thomas Tensorer, Elise Brisebard, Luis Garcia, Aurélie Goyenvalle

**Affiliations:** 1Université Paris-Saclay, UVSQ, Inserm, END-ICAP, 78000 Versailles, France; xaysongkhame-micky.phongsavanh@uvsq.fr (M.P.); olivier.le-coz@uvsq.fr (O.L.C.); luis.garcia@uvsq.fr (L.G.); 2Medical Biology Department, Centre Scientifique de Monaco, 98000 Monaco, Monaco; cgastaldi@centrescientifique.mc; 3LIA BAHN, CSM-UVSQ, 98000 Monaco, Monaco; 4SQY Therapeutics, UVSQ, 78180 Montigny le Bretonneux, France; 5INRAE Oniris, UMR 703 PAnTher, 44300 Nantes, France

**Keywords:** antisense oligonucleotides, exon skipping, RNA, transcript imbalance, valproic acid, histone deacetylase inhibitors, duchenne muscular dystrophy

## Abstract

Duchenne muscular dystrophy (DMD) is a severe genetic disorder characterized by the progressive degeneration of skeletal and cardiac muscles due to the absence of dystrophin. Exon-skipping therapy is among the most promising approaches for treating DMD, with several antisense oligonucleotides (ASO) already approved by the FDA; however, their limited efficacy highlights substantial potential for further improvement. In this study, we evaluate the potential of combining ASO with valproic acid (VPA) to enhance dystrophin expression and improve functional outcomes in a murine model of DMD. Our results indicate that the ASO+VPA treatment significantly increases dystrophin restoration across various muscle tissues, with particularly pronounced effects observed in cardiac muscle, where levels are nearly doubled compared to ASO monotherapy. Additionally, we demonstrate significant improvements in functional outcomes in treated *mdx* mice. Our findings suggest that the combined ASO+VPA therapy holds promise as an effective therapeutic approach to ameliorate muscle function in DMD, warranting further exploration of its mechanistic pathways and long-term benefits.

## 1. Introduction

Duchenne muscular dystrophy (DMD) is a severe X-linked genetic disorder characterized by the progressive degeneration of skeletal and cardiac muscles, primarily caused by mutations in the *DMD* gene that result in the absence of dystrophin, a crucial protein involved in the maintenance of muscle membrane integrity. The lack of dystrophin leads to muscle fiber damage, progressive weakness, loss of ambulation, and ultimately respiratory and cardiac failure, severely impacting the quality of life and life expectancy of affected individuals [1]. Current therapies, such as corticosteroids, offer limited benefits, highlighting the urgent need for innovative curative therapeutic strategies aimed at restoring dystrophin expression and improving muscle function.

Becker muscular dystrophy (BMD) is also associated with mutations in the *DMD* gene, leading to a significantly milder clinical presentation compared to DMD. In contrast with mutations found in DMD patients, which often disrupt the open reading frame, deletions found in BMD patients mostly preserve it, allowing for the production of a partially truncated but still functional dystrophin protein.

Antisense oligonucleotide (ASO)-based therapies, aiming at inducing exon skipping, have emerged as promising strategies for the treatment of DMD. Indeed, the therapeutic antisense-mediated exon-skipping approach for DMD aims to remove one or several exons from the mRNA by utilizing ASOs to block crucial splicing sites during the pre-mRNA splicing process. This technique results in an mRNA with a corrected reading frame, facilitating the expression of a dystrophin similar to that seen in BMD. The US FDA has already approved several ASO-based drugs for the treatment of DMD, including eteplirsen, golodirsen, viltolarsen, and casimersen, which target exons 51, 53, 53, and 45, respectively [2]. A major challenge associated with ASO-mediated exon skipping is effectively delivering ASOs to the targeted tissues [3]. Significant research efforts are currently aimed at enhancing this delivery, particularly through the development of alternative chemical structures or various conjugates such as peptides or antibodies [4]. Many of these novel compounds are presently undergoing, or are soon to undergo, clinical trial evaluations.

Among the alternative chemistries of ASOs that have been investigated, our team has worked on the tricyclo-DNA (tcDNA) [5,6] and shown that conjugating palmitic acid to tcDNA (palm-tcDNA) considerably improves its therapeutic efficacy and delivery to muscle tissues [7,8].

In addition to delivery issues, a lesser-known challenge impacting the efficacy of ASOs is the limited availability of target mRNA. Patients with DMD exhibit a high frequency of transcript imbalance, characterized by non-homogeneous expression of the *DMD* transcript along its length, with the 5′ end being more highly expressed than the 3′ end. Although transcript imbalance in DMD was first described in 1995 [9], its implications for mRNA-based therapies have been highlighted more recently [10,11]. Transcriptional studies have indicated that the levels of dystrophin mRNA are decreased in dystrophic muscles and that this is at least partially attributed to a chromatin structure less conducive to transcription in *mdx* mice compared to wild-type mice [11]. Previous research has demonstrated that valproic acid (VPA) influences gene expression by promoting histone acetylation, which can increase transcriptional activity and improve therapeutic outcomes in models of muscle diseases [12]. Pharmacological inhibition of histone deacetylases has been demonstrated to mitigate fibrosis and enhance muscle regeneration in *mdx* mice, primarily through the upregulation of follistatin. VPA, a branched-chain fatty acid approved by the FDA for the treatment of epilepsy and bipolar disorder, activates the Akt signaling pathway in neurons, thereby promoting their survival. VPA offers several advantages for skeletal muscle, such as improved sarcolemmal integrity, reduced contractures in the hind limbs, and diminished inflammation. Additionally, VPA facilitates hypertrophy and decreases apoptosis in myotubes by engaging the Akt/mTOR/p70S6K signaling cascade.

In a previous study, we showed that the combination of histone deacetylase inhibitors (HDACi) such as givinostat or VPA with exon-skipping therapies over a short period of time (4 weeks) significantly enhances dystrophin expression in *mdx* mice, thereby underscoring the potential of this combined approach [13]. In the present work, we aimed to further investigate the effects of VPA on ASO-mediated exon skipping and functional outcomes in *mdx* mice over an extended treatment period of 12 weeks. We hypothesized that VPA would enhance ASO efficacy in promoting exon skipping and restoring higher dystrophin levels. Our findings demonstrate that VPA treatment significantly increases the accumulation of ASOs in skeletal and cardiac muscle tissues, resulting in elevated levels of exon skipping across various muscle types, particularly in the heart, where dystrophin restoration was found to be twice as high compared to ASO treatment alone. Additionally, assessment of the functional outcomes revealed remarkable improvements in muscle function, as evidenced by the enhanced performances in latency-to-fall tests and grip strength assessments. We also evaluated the safety profile of the combined treatment regimen, focusing on the potential hepatotoxicity and nephrotoxicity, which are common concerns associated with VPA administration. Our assessments indicate an overall favorable safety profile, despite some initial signs of nephrotoxicity and minimal adverse effects, suggesting that the dosing in VPA should be carefully adapted for the therapeutic benefits of the ASO+VPA combination to outweigh potential risks.

## 2. Results

### 2.1. Effect of Valproic Acid on Exon-Skipping Therapy Following a 12-Week Treatment in mdx Mice

To confirm the preliminary data obtained during a previous 4-week study in *mdx* mice [13] and assess the potential of VPA over an extended period of time, we performed a long-term study combining VPA and an ASO aiming at skipping *Dmd* exon 23 in *mdx* mice. During this protocol, mice were weekly injected with tcDNA ASO during 12 weeks (once a week) and treated with VPA 5 times per week during the first month, followed by 3 times per week during the second month and 1 time per week the last month (Figure 1A). The number of VPA injections decreased over time because of its accumulation and to avoid potential toxicities. This dosing regimen was compatible with VPA’s half-life which is estimated between 11 and 20 h in tissues [14], meaning that 99% of the drug is eliminated after 3.2 to 5.8 days.

Two weeks after the last injection of tcDNA, mice were euthanized and tissues were collected. We first assessed the ASO biodistribution in muscles and off target organs and observed a higher amount of ASO in the heart of mice treated with ASO+VPA compared to those treated with ASO alone (Figure 1B). In contrast, a significant decreased accumulation of ASO was observed in the kidneys (*p* < 0.0001 ASO+VPA vs. ASO) (Figure 1B) even though the elimination of ASO in the urines measured during the first 24 h after ASO administration, appeared similar in both treatment groups (Figure 1C). The quantification of exon skipping by real-time TaqMan quantitative PCR revealed higher levels of exon 23 skipping in all muscles examined except in the diaphragm after treatment with ASO+VPA as compared to treatment with ASO alone (Figure 1D). The combined treatment increases skipping efficacy by about 50% on average across all muscles (*p* < 0.0001 ASO+VPA vs. ASO) (Table 1). This effect on RNA was confirmed at the protein level with the quantification of dystrophin restored which appeared to be increased by 70% on average across muscles treated with ASO+VPA compared with ASO alone (*p* < 0.0001 ASO+VPA vs. ASO) (Figure 1E and Table 2). The heart had the highest level of dystrophin restoration, reaching 25% of wild-type (WT) levels (1.8-fold increase compared to ASO alone) while no difference was found between the two treatments in the diaphragm. To determine whether the increase in exon-skipping efficiency and dystrophin restoration was attributable to an elevation in DMD transcript levels following HDAC inhibitor treatment, we then investigated the effect of VPA on the transcript imbalance and quantified the levels of *Dmd* RNA at the various exon junctions (Figure 1F). In contrast with the results obtained in the 4-week study, no significant differences emerged in diaphragm, triceps or heart (Figure 1F and Appendix A for triceps). We could only detect slightly higher levels of transcripts in the heart of ASO+VPA compared to *mdx* or ASO alone, but the trend was not statistically significant.

### 2.2. Functional Impact of VPA Treatment

Dystrophin restoration and its correct localization were also confirmed by immunostaining performed on muscles cryosections (Figure 2A). Quantification of dystrophin staining intensity revealed significantly higher levels in mice treated with VPA+ASO compared to mice treated with ASO alone (Figure 2B, *p* < 0.05 ASO+VPA vs. ASO analyzed by two-way ANOVA) in particular in the hearts of treated mice where the fold change reached 2.3 between the two treatments.

It has been previously demonstrated that the serum level of two fragments of the myofibrillar structural protein myomesin-3 (MYOM3) can be used to evaluate the efficacy of a treatment on the dystrophic pathology [7,15]. We therefore evaluated the level of MYOM3 in serum of treated and control mice after the treatment. Levels of MYOM3 in the serum were significantly decreased by 90% following ASO treatment (*p* < 0.0001) and by 94.2% following ASO+VPA combined therapy (*p* < 0.0001) compared to the untreated mice (Figure 2C). There was no statistical difference between the mice treated with ASO+VPA and the ones treated with ASO only (*p* = 0.2287), which may be explained by the already maximal effect of the ASO treatment (leaving not much space for further improvement).

Previous studies have shown that the effects of HDACi on muscle histopathology is mediated by an upregulation of follistatin, an antagonist of both myostatin and activin A [12,16]. We thus investigated the expression of follistatin in mice treated with ASO alone and detected no upregulation compared to the saline group (*p* = 0.4441 for tibialis anterior and *p* = 0.5575 for diaphragm) (Appendix A). In fact, levels of follistatin appear already upregulated in *mdx* compared to WT muscles (in both tibialis anterior and diaphragm) and the combined treatment ASO+VPA significantly normalizes this level in the diaphragm of treated mice (*p* < 0.01 between saline- and ASO+VPA-treated mice and *p* > 0.9999 between WT and ASO+VPA-treated mice).

To assess the functionality of the restored dystrophin in muscle tissue, a series of functional tests were conducted. Muscle strength was first evaluated using the inverted grid test (Figure 2D), where we measured the latency before the mice fell. WT mice exhibited significantly greater strength, with an average latency of 84.5 s compared to *mdx* mice, which fell after just 12.5 s on average (*p* < 0.001). Remarkably, after 12 weeks of ASO+VPA treatment, the treated mice showed a significant improvement in their latency to fall (*p* < 0.001), achieving similar performance to WT mice (*p* > 0.9999), which was not the case for the ASO group showing a non-statistically significant improvement compared to *mdx* controls (*p* = 0.32).

We further evaluated muscle performance using the wire suspension test (Figure 2E). As expected, there was a significant difference between WT and saline-treated *mdx* mice (*p* < 0.0001). Importantly, a significant improvement was only observed in ASO+VPA-treated mice compared to the saline-treated group (*p* < 0.05).

Next, we measured maximal specific force and resistance to eccentric contraction. Tibialis anterior (TA) muscles from control *mdx* mice demonstrated a 40% reduction in maximal specific force compared to WT muscle (*p* < 0.0001) while treatment with ASO+VPA significantly improved maximal specific force compared to untreated controls (*p*< 0.05) (Figure 2F).

Additionally, TA muscles from untreated *mdx* mice could not maintain tetanic tension, with force dropping to 72% of the initial value after 15 eccentric contractions (*p* < 0.0001) (Figure 2E). In contrast, the combined therapy significantly enhanced resistance to tetanic contractions, with treated muscles retaining 89% of their force following the eccentric contractions (*p* < 0.001 between mdx saline and ASO+VPA; *p* = 0.19 between WT and ASO+VPA).

### 2.3. Impact of VPA Treatment on Central Aspects

Valproic acid is widely used to treat various neurological disorders, such as bipolar disorder and epilepsy. Given this and the known brain comorbidities associated with the lack of dystrophin in the central nervous system (CNS), we examined its effects on the behavior of *mdx* mice. First, we evaluated the unconditioned fear response using the well-established freezing test [17] (Figure 3A). Treatment with the ASO+VPA combination partially restored the fear response, reducing the freezing percentage by half compared to saline-treated mice (*p* < 0.0001). Similarly, vertical activity, which is also an outcome of muscle function, was partially recovered in ASO+VPA-treated mice compared to the saline group (*p* < 0.0001). However, no statistically significant difference was observed in the distance covered during the freezing test in mice receiving the combined therapy (*p* = 0.0576), though a trend towards increased distance was noted. This may be attributed to the anxiety induced by the introduction in a novel environment for emotionally impaired mdx mice, rather than muscle default. Overall, this behavioral test demonstrated a positive effect of the ASO+VPA combination on the fear response, whereas neither VPA nor ASO alone produced similar effects. Indeed, treatment with ASO alone induced only a trend toward improvement but was not statistically significant.

Given the previously demonstrated properties of tcDNA-ASO to cross the blood–brain barrier at low levels (although when administered at higher dose [7]), we sought to determine whether exon skipping occurred in brain regions. We performed a specific RT-PCR using a primer across the skipped junction (exon 22–24) to detect the skipped mRNA in brain samples from *mdx* mice treated with VPA, ASO, or the ASO+VPA combination (Appendix A). Low levels of exon 23 skipped products were detected in the cerebellum and cortex from ASO- and ASO+VPA-treated mice, but not visible in the hippocampus. However, we did not observe any differences between the two treatment groups which could account for the positive behavioral effects observed with the combined therapy. Moreover, we did not detect restoration of Dp427 in any brain tissue (Figure 3D).

Since VPA is a known histone deacetylase inhibitor (HDACi), we investigated whether VPA treatment could affect the 5′-3′ *Dmd* transcripts imbalance in the brain, potentially enhancing transcript availability for exon-skipping therapy. We thus explored the transcript imbalance in the brain, where it had not been previously reported. Quantification of the different exon junctions along the *Dmd* transcript by RT-qPCR confirmed a reduction in 5′-3′ *Dmd* transcripts in *mdx* brain tissues (cortex, hippocampus, and cerebellum) compared to WT controls (Figure 3B). We found a significant reduction in *Dmd* transcripts between the two murine lines (C57/BL10 WT and C57/BL10 mdx) across the three brain regions (*p* = 0.0016 for the cortex, *p* = 0.0002 for the hippocampus, and *p* = 0.0016 for the cerebellum).

We then evaluated whether the combined ASO+VPA therapy could correct the 5′-3′ *Dmd* transcript imbalance. Neither VPA nor the ASO+VPA combination restored this imbalance, as the discrepancy in 5′-3′ DMD transcripts persisted across the exon junctions following treatment with VPA or the combined therapy (Figure 3C).

Finally, we investigated whether the expression of other dystrophin isoforms in the brain, e.g., Dp140 and Dp71 may have been affected by VPA treatment, which could potentially explain the observed improvements in the unconditioned fear response. We quantified the expression of Dp140 and Dp71 in the cortex, hippocampus, and cerebellum of the treated *mdx* mice in comparison to the levels detected in WT mice (Figure 3D). No significant upregulation of these isoforms was found in response to VPA or ASO+VPA treatment compared to PBS-treated *mdx* mice, despite a slight trend for Dp140 in the cortex. Therefore, the improvement in emotional responses seen with the combined treatment cannot be attributed to a modulation of brain dystrophins expression.

### 2.4. Preliminary Safety Assessment of the Combined Treatment of VPA and ASO

Safety pharmacological evaluation is a crucial aspect in the development of a new drug or combination of drugs previously assessed on their own. We used this long-term study to perform a preliminary assessment of the safety profile of the combined therapy, in particular focusing on hepatotoxicity and nephrotoxicity which are the most frequently associated with ASOs. The serum levels of various general biomarkers in mice were analyzed and we found lower levels of transaminases (ALT and AST, biomarkers of liver toxicity) in ASO-treated mice, which is a consequence of the efficacy of the treatment. Apart from these positive effects resulting from the improved dystrophic pathology of treated *mdx* mice, we observed no significant difference in bilirubin, alkaline phosphatase (ALP), albumin, creatinine and urea in the serum of treated mice compared to *mdx* control mice (Figure 4A). The only notable elevation detected was an increase in serum creatine kinase (CK) in VPA-treated mice, a side effect previously reported in human patients receiving VPA treatment [18], which may be attributable to excessively high drug exposure.

We also investigated the histopathological characteristics of the liver and kidneys in treated *mdx* mice and WT controls to assess potential adverse effects of VPA, either administered alone or in combination with ASOs. In the liver, all groups of mice (WT, *mdx* saline, *mdx* ASO, *mdx* VPA, and *mdx* ASO+VPA) showed similar findings, with small, scattered foci of inflammatory cell infiltration, occasionally accompanied by a few necrotic hepatocytes (Figure 4B). Such inflammatory infiltration is a commonly observed background lesion in the liver of mice. The only additional histopathological feature noted was a minimal presence of pigment within Kupffer cells, and less frequently in Ito cells, which was observed in all ASO-treated animals (both ASO alone and in combination with VPA). This minor pigment deposition, unaccompanied by any cellular reaction or lesions, is typically considered biologically insignificant. In the kidney, when all WT, mdx saline-, and mdx ASO-treated mice displayed no lesion or only unspecific sporadic changes (proteinaceous casts, altered glomeruli, basophilic tubules), all mice treated with VPA, alone or combined with ASO, presented areas of renal injury characterized by histopathologic changes ranging from tubular degeneration and regeneration associated with various degree of interstitial inflammation and intraluminal proteinaceous cast, to tubular atrophy and interstitial fibrosis. The lesions tended to be more pronounced in the ASO+VPA group in comparison with the VPA group. Even though, the renal histopathological lesions were overall minimal to mild in extension and severity, they were VPA compound related and signs of a renal progressive injury and should be considered a significant adverse effect of the VPA treatment.

In order to investigate further the potential renal toxicity of the VPA treatment and the combined therapy, several biomarkers were examined in the urines of the treated mice. Urines were collected during 24 h following the last injection of ASO and levels of total protein and albumin were measured and normalized to creatinine levels. No changes in normalized total protein levels were found after treatment (Figure 4C). Urinary albumin levels are elevated in *mdx* mice compared to the WT ones and these levels were normalized after ASO treatment (*p* < 0.05) but slightly increased by VPA (although not significantly, *p* = 0.0583).

We also assessed urinary kidney injury biomarkers (KIB) in treated *mdx* mice and found no significant differences across all evaluated markers, including beta2-microglobuline (B2M), renin, kidney injury molecule-1 (KIM-1), interferon gamma-inducible protein 10 (IP-10), vascular endothelial growth factor (VEGF), cystatin C, epidermal growth factor (EGF), clusterin, and osteopontin (OPN). A notable exception was the elevated levels of lipocalin-2/NGAL detected in mice treated with VPA alone; however, this change was not statistically significant and due to the increased level detected in a single animal.

## 3. Discussion

Duchenne muscular dystrophy poses a major challenge in neuromuscular medicine notably due to the extensive size of the affected tissues, since skeletal and cardiac muscles represent a significant portion of the body. While ASO-mediated exon skipping is a promising therapy, its efficacy is hindered by challenges like poor ASO delivery but also the limited availability of dystrophin pre-mRNA, exacerbated by a 5′-3′ imbalance. Elevated H3K9me3 levels and increased modifications at the *Dmd* locus in *mdx* mice have suggested reduced transcription due to restrictive chromatin conformation [11]. We thus propose that HDAC inhibitors could enhance histone acetylation, improve transcription, and increase pre-mRNA availability for exon skipping. Our study aimed to evaluate the combined effects of an ASO targeting the *Dmd* exon 23 and the HDACi VPA on dystrophin restoration and functional outcomes in *mdx* mice. Our findings reveal that the combination of ASO and VPA resulted in a substantial increase in dystrophin restoration across all muscle tissues examined (+70% on average), with particularly noteworthy effects observed in the heart, where dystrophin levels were restored to approximately two times greater than those achieved with ASO treatment alone. This enhancement highlights the potential of the ASO and VPA combination to increase dystrophin levels across different muscle types, which is crucial for effective treatment. These results confirm the preliminary results obtained during a short-term (4 weeks) study [13] but more importantly demonstrate a significant improvement in functional outcomes with the combined therapy while the ASO therapy alone often did not reach statistically significant differences. We intentionally selected a lower dose of ASO than used in previous studies on tcDNA-ASO [7] to (i) allow room for improvement with combined VPA treatment and (ii) simulate potential clinical scenarios where ASO therapies, such as FDA-approved PMOs, achieve dystrophin restoration levels insufficient for meaningful clinical outcomes. For instance, the approval of PMO-ASO drugs for DMD (eteplirsen, golodirsen, viltolarsen, and casimersen) was based on safety and modest increases in dystrophin expression in muscle biopsies, reaching levels of up to ~5.9% after exon 53 skipping [19]. Although long-term eteplirsen treatment has now shown some benefits, such as delayed loss of ambulation and slower pulmonary decline compared to natural history cohorts [20], the overall efficacy of these exon-skipping strategies remains limited. This underscores the potential value of approaches like HDACi co-treatment to enhance therapeutic outcomes.

Yet, when we assessed the transcript imbalance after the various treatments, we did not observe a correction of the 5′-3′ difference in transcript levels and detected at best a tendency for higher overall levels of *Dmd* transcripts across all exon junctions in heart. This suggests that higher dystrophin restoration and improved functional outcomes are not resulting from a correction of the imbalance as we initially hypothesized. In fact, the positive impact of VPA co-treatment is already detected at the exon-skipping levels, although to a lesser extent than protein level (+50% on average across muscles).

Pharmacological blockade of histone deacetylases has previously been reported to reduce fibrosis and promote compensatory regeneration in *mdx* skeletal muscle notably through the upregulation of follistatin [12,21,22]. However, we did not observe the upregulation of follistatin here in mice treated with VPA (with or without ASO) suggesting a different mechanism underlying the increased dystrophin expression and improved functional benefits.

The higher levels of dystrophin in skeletal and cardiac muscles likely play a crucial role in improving muscle integrity and function, which contributes directly to physical performance. The stabilization of muscle membranes during contraction due to increased dystrophin levels can lead to improved force generation and reduced muscle damage, facilitating better overall muscle performance. In addition to its role as an HDAC inhibitor that modulates protein levels, VPA also affects ion channel function in healthy individuals [23,24,25]. Therefore, VPA may reduce susceptibility to contraction-induced functional loss by maintaining muscle excitability through ion channels, as the decrease in force following eccentric contractions is also linked to impaired membrane excitability [26,27]. Recent studies have indeed demonstrated that the force drop following eccentric contractions is significantly reduced after 7 days of VPA treatment in *mdx* mice [28].

The positive impact of the VPA co-treatment may be attributed to its multiple biological effects on skeletal muscles including decreased fibrosis, damage and inflammation as well as increased sarcolemmal integrity [16]. Given that VPA treatment alone has previously been associated with such outcomes, it makes sense that combining it with ASO therapy, which restores dystrophin expression and produces similar effects, would result in a synergistic benefit.

Moreover, HDACi were previously shown to stimulate myogenesis in various cell types, including pluripotent stem cells [29] and fibro-adipogenic progenitors (FAPs) [30]. Additional studies have revealed that exposure to HDACi promotes the formation and release of pro-regenerative and antifibrotic extra-cellular vesicles (EVs) from FAPs of DMD muscles [31]. Thus, the modified microenvironment induced by HDACi may enhance overall muscle tissue health, which in turn could facilitate improved uptake of ASOs and/or promote the recovery of dystrophin protein induced by ASOs.

When evaluating the *Dmd* transcript imbalance in this study, we found no correction following ASO treatment, consistent with findings by Spitali et al., who observed no improvement in transcript imbalance rates after PMO-ASO treatment in *mdx* mice [10]. However, these results contrast with recent work by Rossi and colleagues [32], who demonstrated for the first time that golodirsen (a PMO-ASO targeting exon 53) restored transcript imbalance in muscle cells from DMD patients. Specifically, golodirsen-treated cultures expressed significantly higher levels of *DMD* transcripts compared to untreated patient cultures, though the imbalance was not fully corrected to levels seen in healthy controls. This discrepancy may stem from differences in the biological context of the models used: their in vitro cellular system versus our in vivo model. In addition to the inherent complexity differences between in vivo and in vitro systems, the exon-skipping levels observed in patient cells were much higher than those measured in vivo, likely facilitated by the use of the transfection agent EndoPorter, which bypasses ASO delivery challenges. Consequently, the lower skipping levels achieved in vivo in *mdx* mice may explain the lack of transcript imbalance correction in our system, although the levels observed in our study, ranging from 7% to 25%, are already considerably higher than those typically reported in clinical trials. Another contributing factor could be the distinct mutations and their localizations: the *mdx* mouse model harbors a nonsense mutation in *Dmd* exon 23, whereas the DMD patient cells had more distal deletions spanning *DMD* exons 45–52.

Despite these differences, both studies underscore the importance of considering the 5′-3′ transcript imbalance phenomenon in ASO therapy for DMD. Overlooking this phenomenon could lead to underestimations of skipping efficiency and transcript expression, as well as inaccurate predictions of dystrophin restoration in humans, regardless of the ASO backbone used.

Overall, our study demonstrates that the combined treatment of ASOs and VPA not only significantly enhances dystrophin restoration across various muscle tissues but also leads to notable improvements in functional outcomes. Future studies should aim to evaluate the impact of VPA co-treatment on cardiac function, particularly given the observed improvements in dystrophin restoration in the heart, to fully elucidate the therapeutic potential of this combined approach. The substantial increase in dystrophin levels in skeletal muscles and the heart, averaging +70%, highlights the efficacy of this combination therapy. However, while the safety profile was generally favorable, early signs of nephrotoxicity and minimal adverse effects observed through detailed histopathology underscore the need for careful dose optimization to ensure the therapeutic benefits outweigh potential risks. To mitigate the observed nephrotoxicity, future studies should explore dose optimization strategies, including lower or fractionated dosing of VPA, which may reduce renal stress while maintaining therapeutic efficacy. Adjusting dosing schedules, such as intermittent dosing or using combination treatments with protective agents, may also reduce cumulative toxicity. These approaches will be crucial in refining the therapeutic potential of HDACi and ASO co-treatments for DMD. Moreover, this study focused on a single ASO targeting exon 23, and further studies are needed to determine whether VPA enhances exon-skipping efficiency across other ASO sequences and in different DMD mutations. While VPA is a widely used HDAC inhibitor, it has broad epigenetic effects, and potential off-target actions cannot be excluded. Future investigations should explore alternative HDAC inhibitors with greater specificity or improved safety profiles to optimize therapeutic outcomes while minimizing unintended effects.

Evaluating the combination treatment in larger animal models, such as canine models of DMD, would also provide valuable translational insights into its efficacy and safety in a more clinically relevant setting. Furthermore, given the broad impact of HDAC inhibitors on gene regulation, it would be interesting to investigate whether VPA or other HDAC inhibitors could enhance the efficacy of additional gene therapies for DMD, such as gene replacement with microdystrophin.

While our initial hypothesis regarding the correction of the 5′-3′ transcript imbalance was not validated, the observed improvements in dystrophin restoration and functional outcomes suggest that alternative mechanisms, such as enhanced myogenesis or maintenance of muscle excitability, may underlie the beneficial effects of VPA. Further investigations are warranted to unravel the molecular mechanisms underlying the synergistic effects of HDAC inhibitors like VPA in this combined approach and to refine strategies that maximize clinical benefits for DMD patients and potentially other neuromuscular disorders. In summary, our findings provide compelling evidence supporting VPA as an effective adjuvant to ASO therapy, paving the way for future studies into HDAC inhibitors in exon-skipping therapies and offering new hope for improving treatments for individuals affected by DMD.

## 4. Materials and Methods

### 4.1. Antisense Oligonucleotides and Animal Experiments

Animal procedures were carried out in compliance with both national and European regulations and were approved by the French government (Ministère de l’Enseignement Supérieur et de la Recherche, Autorisation APAFiS #6518). Mdx (C57BL/10ScSc-Dmdmdx/J) mice were bred in our animal facility at the Plateforme 2Care, UFR des Sciences de la Santé, Université de Versailles Saint-Quentin, and were maintained under a standard 12 h light/dark cycle with ad libitum access to food and water. Mice were weaned at 4–5 weeks postnatally, with 2–5 mice housed per cage. TcDNA-ASO targeting the donor splice site of exon 23 of the mouse dystrophin pre-mRNA [6] was synthesized by SQY Therapeutics (Montigny-le-Bretonneux, France). Palmitic acid was conjugated to the 5′ end of tcDNA-PO using a C6-amino linker and a phosphorothioate bond, as previously described [7].

Groups of 8–10-week-old *mdx* mice were injected intravenously with 30 mg/kg/wk of the tcDNA-ASO (one intravenous injection per week under general anesthesia using 2% isoflurane) together with valproic acid (Santa Cruz Biotechnology, Dallas, TX, USA, dissolved in PBS, used at a final concentration of 500 mg/kg/day) or saline for 12 weeks (n = 8 mice per group). Valproic acid was administered intraperitoneally 5 times per week for the first month, as previously described [16] and, due to the long duration of this study, mice were then treated with VPA 3 times per week during the second month, and once per week during the final month. Age-matched *mdx* groups receiving an equivalent volume of sterile saline were used as controls, and C57BL/10 mice were included as wild-type controls.

Animals were euthanized 2 weeks after the final ASO injection. Muscles and brain tissues were harvested, snap-frozen in liquid nitrogen-cooled isopentane, and stored at −80 °C for subsequent analysis.

### 4.2. Serum and Urine Analysis

Blood samples were collected at the end of the treatment for myomesin-3 (MYOM-3) and biochemical analysis. Serum levels of alanine aminotransferase (ALT), aspartate aminotransferase (AST), alkaline phosphatase (ALP), bilirubin, creatinine, urea, and albumin were analyzed by the pathology laboratory at the Mary Lyon Centre, Medical Research Council, Harwell, Oxfordshire, UK.

Urine samples were collected over a 24 h period using metabolic cages, and the concentrations of creatinine and total protein were measured, following previously established protocols [33].

### 4.3. Histopathological Profile of Liver and Kidneys

To evaluate the safety of valproic acid, liver and kidney tissues were collected at the end of the 12-week study protocol (two weeks after the final dose), fixed in 10% neutral buffered formalin, and embedded in paraffin. Four-micron thick sections were prepared and stained with hematoxylin–eosin–saffron (HES) for standard histopathological analysis. The evaluation was conducted by a veterinary pathologist who was blinded to the treatment groups.

### 4.4. ASO Quantification by Fluorescent Hybridization Assay

Tissues were homogenized using the Precellys 24 (Bertin Instruments, Montigny le Bretonneux, France) with lysis buffer (100 mmol/L Tris–HCl, pH 8.5, 200 mmol/L NaCl, 5 mmol/L EDTA, 0.2% sodium dodecyl sulfate) containing 2 mg/mL of proteinase K (Thermo Fisher Scientific, Montigny le Bretonneux, France) at a final concentration of 50 mg of tissue per ml of buffer. The homogenates were then incubated overnight at 55 °C in a hybridization oven. Following a 15 min centrifugation at 7000× *g* (Sorval ST 8R centrifuge, 75005719 rotor), the supernatant was collected for analysis. ASO quantification was carried out using a hybridization assay with a molecular beacon probe, as previously described [34]. The concentration of tcDNA in tissues was determined by comparing the fluorescence signals to a standard curve generated from known quantities of tcDNA dissolved in tissue lysates from mock-injected animals.

### 4.5. RNA Analysis

Total RNA was extracted from snap-frozen muscle tissues using TRIzol reagent following the manufacturer’s protocol (Thermo Fisher Scientific, Montigny le Bretonneux, France. Exon 23 skipping was quantified using real-time quantitative PCR with TaqMan assays targeting the exon 23–24 and exon 22–24 junctions, as previously described [7]. The copy numbers for the skipped product (exon 22–24) and the unskipped product (exon 23–24) were calculated using standards Ex20–26 and Ex20–26Delta23, respectively, which were gBlocks gene fragments from Integrated DNA Technology. Exon 23 skipping was expressed as a percentage of total *Dmd* transcripts, calculated by adding the copy numbers of exon 22–23 and exon 22–24.

Exon 23 skipping was also visualized on gel after RT-PCR using a forward primer in exon 20 (ex20F: 5′-CCCAGTCTACCACCCTATCAGAGC-3′) and a reverse primer across the exon 22–24 junction (Ex22–24R: 5′-TTATGTGATTCTGTAATTTC-3′).

Quantification of *Dmd* transcripts at various exon-exon junctions was performed similarly, with probes targeting the ex4–5 (Mm.PT.58.41636025), ex49–50 (Mm.PT.58.6233636), ex58–59 (Mm.PT.58.43613256), ex65–66 (Mm.PT.58.42993407), and GAPDH (Mm.PT.39a.1) junctions (Integrated DNA Technology, Coralville, IA, USA). Absolute quantification was determined using corresponding standards (gBlocks gene fragments from Integrated DNA Technology).

Follistatin quantification was conducted by real-time quantitative PCR using TaqMan assays (Mm.PT.58.11399784 from Integrated DNA Technology, Coralville, IA, USA), with delta-delta CT analysis using GAPDH for normalization.

### 4.6. Western Blot in Muscle Tissues

Protein lysates were prepared from muscle sections collected during cryosectioning using the Precellys 24 (Bertin Instruments, France) in RIPA buffer (Thermo Fisher Scientific, Montigny le Bretonneux, France) supplemented with SDS powder (5% final concentration, Bio-Rad, Marnes-la-Coquette, France) and protease inhibitor cocktail (Thermo Fisher Scientific, Montigny le Bretonneux, France). The protein extracts were denatured by heating at 100 °C for 3 min, followed by centrifugation at 13,000 rpm for 10 min at 10 °C. The supernatants were collected, and the total protein concentration was quantified using the BCA Protein Assay Kit (Thermo Fisher Scientific, Montigny le Bretonneux, France). A total of 25 μg of protein was loaded onto NuPAGE 3–8% Tris-Acetate Protein gels (Invitrogen) according to the manufacturer’s instructions. Dystrophin was detected using the iBind™ Flex Western Device (Thermo Fisher Scientific, Montigny le Bretonneux, France) and probed with the primary monoclonal antibody NCL-DYS1 (Novocastra, Newcastle, UK, dilution 1/200) and hVin-1 (Sigma, dilution 1/4000). After incubation with a goat anti-mouse secondary antibody (IRDye 800CW goat anti-mouse IgG, Li-Cor, Bad Homburg, Germany, dilution 1/2000), protein bands were visualized with the Odyssey CLx system (Li-Cor, Bad Homburg, Germany). Quantification of dystrophin expression was performed using Empiria Studio 2.3 software (Li-Cor, Bad Homburg, Germany), with data normalized against a standard curve generated from pooled lysates of C57BL10 (wild-type) and *mdx* control tissues.

For myomesin-3 detection, mouse serum was diluted 1:20 and loaded onto 3–8% Criterion™ XT Tris-Acetate Protein Gel (Biorad, Marnes-la-Coquette, France) according to the manufacturer’s protocol. Myomesin-3 was detected by probing the membrane with a primary rabbit polyclonal antibody against MYOM3 (Proteintech, Manchester, UK), followed by a secondary goat anti-rabbit antibody (IRDye 800CW goat anti-rabbit IgG, Li-Cor, Bad Homburg, Germany). Bands were visualized with the Odyssey Imaging System (Biosciences, Lincoln, NE, USA). The intensity of the signals in treated samples was quantified and normalized to the PBS control group using Image Studio 2.1 software (Li-Cor, Bad Homburg, Germany).

### 4.7. Western Blot in Brain Tissues

Protein extracts were prepared from brain tissues using RIPA lysis and extraction buffers (Thermo Fisher Scientific, Rockford, IL, USA), supplemented with SDS powder (5% final concentration) (Bio-Rad, Marnes-la-Coquette, France). The total protein concentration was assessed using the BCA Protein Assay Kit (Thermo Fisher Scientific, Rockford, IL, USA). Samples were denatured by heating at 100 °C for 3 min, and 25 μg of protein was loaded onto NuPAGE 3–8% Tris-Acetate Protein gels (Thermo Fisher Scientific, Rockford, IL, USA), according to the manufacturer’s protocol. Dystrophin was detected by incubating the membrane with the AB154168 rabbit primary monoclonal antibody (AB154168, Abcam, Paris, France), and vinculin was used as an internal control with the hVin-1 primary antibody (Sigma, St. Louis, MI, USA). Secondary antibodies included IRDye 800CW goat anti-mouse IgG (Li-Cor, Bad Homburg, Germany) for vinculin detection and IRDye 700CW goat anti-rabbit IgG (Li-Cor, Bad Homburg, Germany) for dystrophin detection. Protein bands were visualized using the Odyssey CLx imaging system (Li-Cor, Bad Homburg, Germany). Densitometric analysis was performed using Empiria Studio software version 3.0 (Li-Cor, Bad Homburg, Germany), with quantification normalized to the vinculin internal control. A standard curve was created for each brain structure using a mixture of WT and mdx control lysates, representing known dystrophin percentages (0%, 5%, 10%, and 20% of corresponding WT tissues).

### 4.8. Immunohistochemistry Analysis

Ten-micron sections were prepared from triceps and heart tissues and examined for dystrophin expression using a rabbit polyclonal antibody against dystrophin (dilution 1:500; cat. number RB-9024-P, Thermo Fisher Scientific, Montigny le Bretonneux, France). Dystrophin was detected with donkey anti-rabbit IgG conjugated to Alexa 594 (dilution 1:400; Jackson Immuno Research, Cambridge, UK). Control sections, where the primary antibody was omitted, exhibited no specific staining. Images were captured at consistent locations and exposure times using a Zeiss Axio Imager, equipped with an Orkan camera (Hamamatsu) and AxioVision 4.7 software. Image analysis was performed using ImageJ 1.52 software.

### 4.9. Restraint-Induced Unconditioned Fear

Mice were gently restrained by holding the scruff and back skin between the thumb and index fingers, securing the tail between the third and little fingers, and positioning the animal upside down with its ventral side facing the experimenter. After 15 s of restraint, the mouse was placed into a novel cage with clean sawdust and monitored for 5 min under dim lighting using Any-maze 7.4 software (Stoelting). The unconditioned fear response to this stress was characterized by periods of tonic immobility (freezing), which were quantified during the 5 min recording session. Complete cessation of movement, except for respiration, was considered as freezing. The percentage of time spent freezing was calculated for group comparisons. The distance is calculated automatically by Any-maze software using tracking option and the vertical activity is measured manually by the experimenter considering every time the mouse is standing on their posterior paws.

### 4.10. Susceptibility to Contraction-Induced Loss of Function

Maximal forces before and after eccentric contractions were assessed by measuring the in situ contraction of the tibialis anterior (TA) muscle in response to nerve stimulation, as previously described [7,27]. The susceptibility of *mdx* mice to contraction-induced functional loss was evaluated by measuring the immediate drop in isometric force following eccentric contractions. Fifteen eccentric contractions were applied, each separated by a 45 s rest period. Maximal isometric force was measured after each contraction and expressed as a percentage of the initial maximal force. Additionally, absolute maximal eccentric force was recorded during the first eccentric contraction, along with an index of muscle strain. Following these measurements, the mice were euthanized by cervical dislocation.

### 4.11. The Inverted Grid Test

Mice were individually placed on a horizontal wire grid (size typically 25 × 25 cm) which was then inverted to a vertical position at 50 cm height above an experimental cage with sawdust, causing the mouse to be upside down. The time the mouse was able to remain suspended on the grid or the time until it fell off was recorded. Mice were given a maximum of 2 min to remain on the grid. A failure to maintain grip within this time frame was considered as a loss of grip strength. The duration of time the mouse remained on the grid was used as an indicator of muscular strength and coordination. Differences between the groups were analyzed using one-way ANOVA with post hoc comparisons.

### 4.12. The Wire Suspension Test

Mice were positioned on a horizontal wire, and their ability to grip the wire was assessed by suspending them from the wire using their forelimbs and tail. The wire was suspended 50 cm above the surface. The time the mouse maintained its grip on the wire without falling was recorded. The test was conducted for a maximum of 3 min. If the mouse lost its grip and fell, the test was ended, and the time to fall was recorded.

The duration of suspension was used as an indicator of muscle strength and endurance. Statistical analysis, using one-way ANOVA, was conducted to compare the performance across different groups.

### 4.13. Statistical Analysis

All in vivo data were analyzed using GraphPad Prism10 software (San Diego, CA, USA) and are presented as means ± S.E.M. The “n” represents the number of mice per group. Group comparisons were conducted using one-way and two-way analyses of variance (ANOVA), with repeated measures when appropriate (e.g., comparing effects across different exon junctions or muscle tissues), followed by post hoc Dunnett’s or Sidak’s multiple comparisons tests as needed. For the comparison of overall treatment effects across different tissues or exon junctions, a two-way ANOVA was applied, and the *p* for the treatment effect is reported in the figure legend. The Kruskal–Wallis test was used for comparisons of groups that did not follow a normal distribution (assessed using the Shapiro–Wilk test).

## Figures and Tables

**Figure 1 ijms-26-02583-f001:**
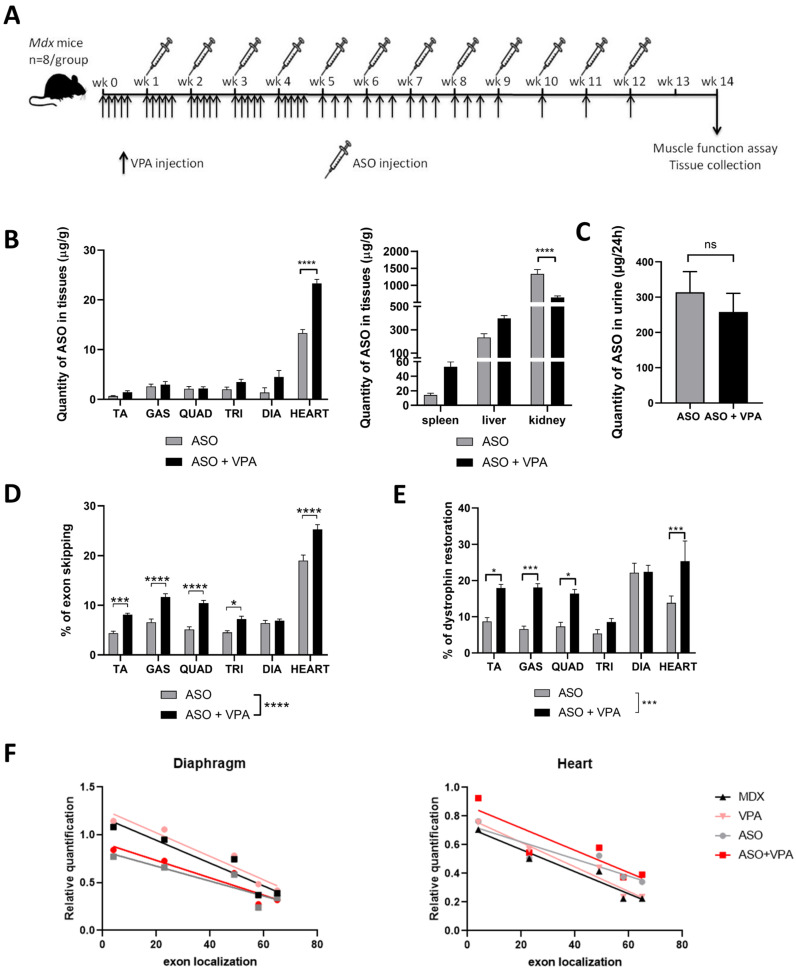
Long-term treatment with VPA increases exon-skipping efficacy. (**A**) Schematic representation of the treatment of *mdx* mice with ASO combined with valproic acid (VPA). (**B**) Effect of VPA on tcDNA-ASO distribution in various tissues. (**C**) Quantification of tcDNA-ASO in urines of treated mdx mice; urines were collected over a period of 24 h in metabolic cages. (**D**) Quantification of exon 23 skipping levels by TaqMan RT-qPCR in *mdx* mice treated with ASO or ASO+VPA. (**E**) Dystrophin restoration quantified by Western blot in *mdx* mice treated with ASO or ASO+VPA. (**F**) Effect of the combination VPA+ASO on *Dmd* transcript imbalance in diaphragm, and heart analyzed by TaqMan qPCR at different exon junctions. TA: tibialis anterior, GAS: gastrocnemius, QUAD: quadriceps, TRI: triceps and DIA: diaphragm. Results are expressed as the mean ± SEM; n = 6–8 mice per group, * *p* < 0.05, *** *p* < 0.001, **** *p* < 0.0001 and ns: non statistically significant analyzed by two-way ANOVA.

**Figure 2 ijms-26-02583-f002:**
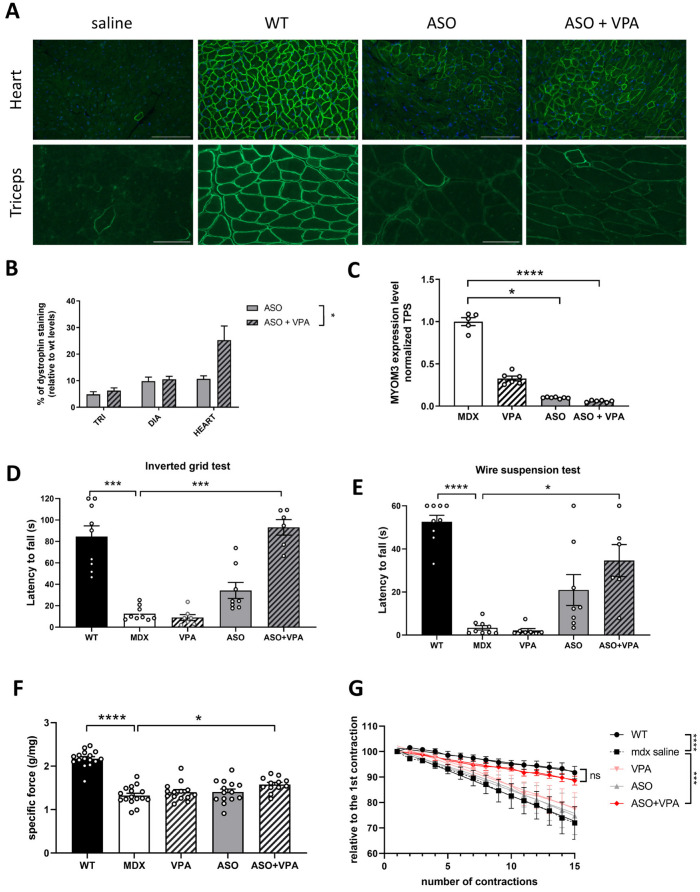
Functional recovery following the combined therapy ASO+VPA. (**A**) Detection of dystrophin protein (green staining) by immunostaining on transverse sections of muscle tissues (triceps and heart) from WT and *mdx* mice treated with saline, ASO, VPA or ASO+VPA. Nuclei are labelled with DAPI (blue staining). Scale bar, 100 µm. (**B**) Quantification of the dystrophin intensity staining in heart and triceps; n = 4 mice per group. (**C**) Myomesin-3 levels in serum of treated mice detected by Western blot; n = 5–7 mice per group. (**D**) Latency to fall in seconds in the inverted grid test; n = 6–9 mice per group. (**E**) Latency to fall in seconds in the wire test; n = 6–9 mice per group. (**F**) Maximal specific force in mg/g measured from the two tibialis anterior muscles of each mouse; n = 6–8 mice per group. (**G**) percentage of force drop following a series of 15 eccentric contractions measured on semi-isolated tibialis anterior muscles from treated *mdx* mice; n = 6–8 mice per group. Results are expressed as the mean ± SEM; * *p* < 0.05, *** *p* < 0.001, **** *p* < 0.0001 and ns: non statistically significant analyzed by two-way ANOVA.

**Figure 3 ijms-26-02583-f003:**
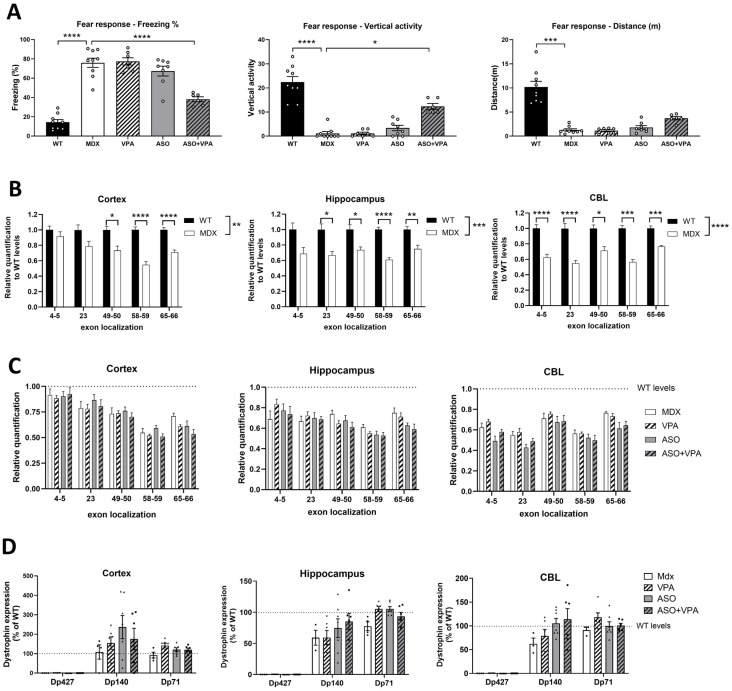
Impact of long-term treatment with VPA in the brain. (**A**) Unconditioned fear response expressed as percentage of freezing (left panel), vertical activity (middle panel) or distance traveled (right panel) during a 5 min period of observation following a brief scruff restraint (15 s) (mean ± SEM; * *p* < 0.05; *** *p* < 0.005; **** *p* < 0.0001 analyzed by one-way ANOVA followed by Sidak post hoc tests). (**B**) Relative expression of *Dmd* transcript levels in *mdx* compared to WT levels quantified in cortex, hippocampus and cerebellum (CBL) at various exon–exon junctions along the *Dmd* gene (n = 8 per group) (* *p* < 0.05; ** *p* < 0.01 in cortex, *** *p* < 0.0001 in hippocampus and **** *p* < 0.0001 in cerebellum). (**C**) Relative expression of *Dmd* transcript levels in *mdx* mice treated with PBS (MDX), VPA, ASO, or the combination of ASO+VPA at various exon–exon junctions along the *Dmd* gene (n = 8 per group). (**D**) Quantification of the different dystrophins expression (Dp427, Dp140 and Dp71) in brain tissues (in cortex, hippocampus and cerebellum (CBL)) treated with VPA, ASO, or the combined therapy ASO+VPA by Western blot. The expression of each dystrophin is expressed as a percentage of levels detected in WT mice. Results are expressed as the mean ± SEM.

**Figure 4 ijms-26-02583-f004:**
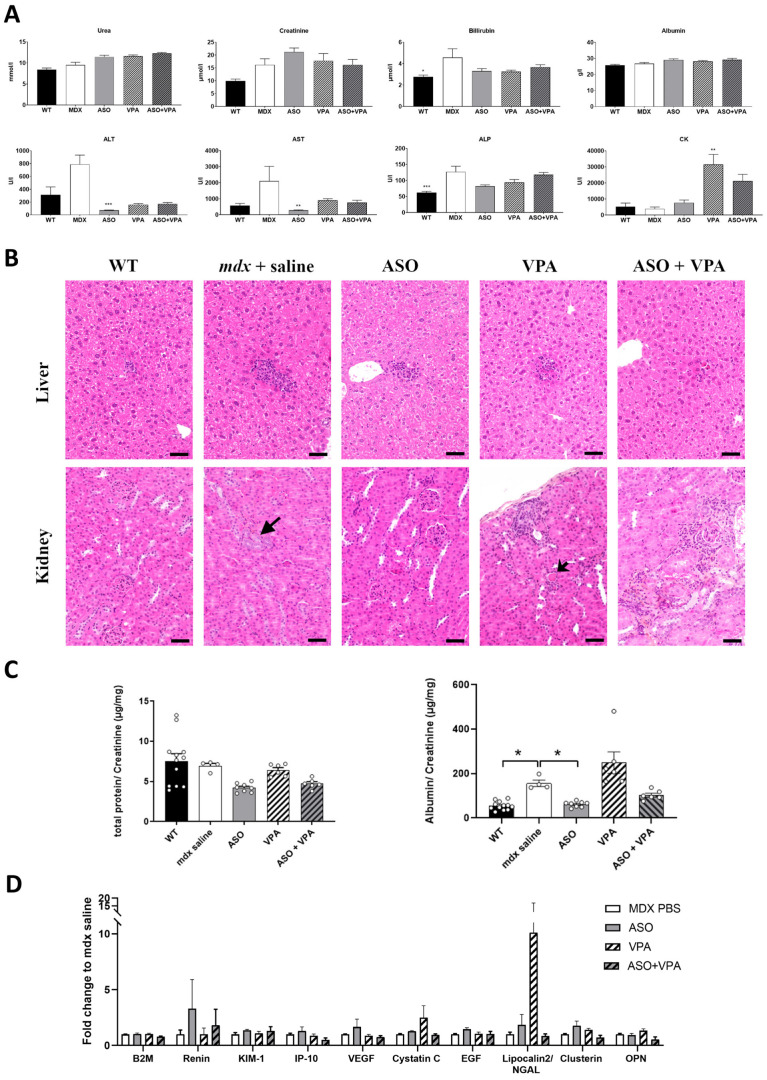
Evaluation of the safety profile of the combined ASO+VPA therapy. (**A**) Quantification of general toxicity biomarkers in the serum: creatinine, urea, albumin, aspartate aminotransferase (AST), alanine aminotransferase (ALT), alkaline phosphatase (ALP) and bilirubin; n = 7 mice per group, * *p* < 0.05, **, *p* < 0.01, and *** *p* < 0.001 compared to *mdx* saline, analyzed by the Kruskal–Wallis test. (**B**) Histopathological analysis from liver and kidney sections from WT and *mdx* mice treated with saline solution, ASO, VPA or ASO+VPA, stained with hematoxylin–eosin–saffron. In liver (upper panel), small foci of inflammatory cell infiltration were scattered in the hepatic parenchyma of all mice in every group. In kidney (lower panel), no lesions or only sporadic changes were observed in WT mice and *mdx* saline- and *mdx-*ASO-treated mice, whereas all VPA and ASO+VPA mice presented histopathological changes ranging from tubular degeneration/regeneration associated with various degree of proteinaceous casts (short arrow) and interstitial inflammation to tubular atrophy and loss and interstitial fibrosis. These changes included basophilic tubules, an early unspecific change that is a relatively common finding in aging mice and was present focally in one *mdx* saline mouse (arrow). Scale bar = 50 µm. (**C**) Quantification of total protein/creatinine (left panel) and albumin/creatinine (right panel) in urines of WT and *mdx* mice treated with saline solution, ASO, VPA or ASO+VPA. * *p* < 0.05 compared to *mdx* saline, analyzed by the Kruskal–Wallis test. (**D**) Quantification of kidney injury biomarkers in the urines of treated mice; n = 7 mice per group, two-way ANOVA analyses. Results are expressed as the mean ± SEM.

**Table 1 ijms-26-02583-t001:** Exon-skipping efficacy following VPA treatment in *mdx* mice. Fold changes are calculated from the levels of exon 23 skipping quantified by TaqMan qPCR obtained in the muscles co-treated with ASO and VPA compared to the same muscles treated with ASO alone.

Exon-Skipping Efficacy (%)	TA	GAS	QUAD	TRI	DIA	HEART	Average
ASO	4.4	6.6	5.1	4.5	6.4	19.0	7.7
ASO+VPA	8.1	11.7	10.4	7.2	6.9	25.3	11.6
**Fold change**	**1.8**	**1.8**	**2.0**	**1.6**	**1.1**	**1.3**	**1.5**

**Table 2 ijms-26-02583-t002:** Protein restoration efficacy following VPA treatment in *mdx* mice. Levels of dystrophin restoration quantified by WB in the muscles treated with ASO or ASO+VPA and fold changes calculated between the two treatments.

Protein Restoration (% of WT)	TA	GAS	QUAD	TRI	DIA	HEART	Average
ASO	8.7	6.6	7.3	5.4	22.2	13.9	10.7
ASO+VPA	17.9	18.1	16.4	8.6	22.4	25.6	18.1
**Fold change**	**2.1**	**2.8**	**2.2**	**1.6**	**1.0**	**1.8**	**1.7**

## Data Availability

The data presented in this study are available in this article and in the Appendix A.

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
