# Peer review of "Valproic Acid Improves Antisense-Mediated Exon-Skipping Efficacy in mdx Mice"

_ijms, 2025, doi:10.3390/ijms26062583_

Round 1

Reviewer 1 Report

Comments and Suggestions for Authors

Lane 49: ''Antisence oligonucleotide (ASO).....'' - I think this should be a new paragraph;

Lane 85: ''In a previous study.....'' - I think this should be a new paragraph;

Lane 86: - concerning the abbreviation HDAC - perhaps this means ''histone deacetylases inhibitors''. The authors mention this somewhere in the section Results, but this abbreviation should be specified in section Introduction;

Lane 340-341, legend of fig. 4B: the size of the scale bar is missing;

Fig. 4B: The meaning of the arrows on the kidney photos should be described in the legend. In addition, the authors mention in the text necrotic foci in the liver samples - perhaps they should indicate them on the pictures.

Reviewer 2 Report

Comments and Suggestions for Authors

- Although the study shows that VPA improves the effectiveness of ASO-mediated exon skipping, it is yet unknown how exactly VPA accomplishes this. As a histone deacetylase inhibitor (HDACi), the authors speculate that VPA may enhance transcription and raise pre-mRNA availability for exon skipping. A crucial component of the original hypothesis, the 5'-3' transcript imbalance, was not significantly corrected in the data.

- Even though there are a number of functional assessments in the study, the functional gains are rather slight. To offer a more thorough assessment of the therapeutic benefits, the authors want to think about using extra functional assays, such as rotarod, treadmill running, or more in-depth evaluations of cardiac function.

- According to the study, mice given VPA, especially those in the ASO+VPA group, show early symptoms of nephrotoxicity. Nephrotoxicity may restrict the combination therapy's clinical usefulness, which is a serious problem. The authors ought to talk about possible ways to lessen this toxicity, such changing the dosage, using less harmful HDAC inhibitors, or using different dosing schedules.

- Although the results are not consistent, the ASO+VPA treatment does show some improvement in the behavioral tests (such as the unconditioned fear response). To properly evaluate how the therapy affects CNS function, the authors ought to think about incorporating more sensitive behavioral tests.

- More thorough information on the transcript imbalance, ASO distribution, and histology findings should be included in the main publication or as extra figures, even though the study cites supplementary materials. This would improve the study's repeatability and transparency.

- The study's limitations, such as the use of a single ASO that targets exon 23 and the possibility of off-target effects of VPA, should be specifically addressed by the authors.

- The authors ought to specify certain avenues for future study, such examining alternative HDAC inhibitors, evaluating the combo treatment in larger animal models, or examining how VPA might improve other kinds of gene treatments for DMD.

Reviewer 3 Report

Comments and Suggestions for Authors

This is a high quality study on the exon-skipping effects of both anti-sense oligos and VPA. The idea quite novel and there is sufficient data to support the conclusions of the paper. The authors may want to discuss if further tests of ASOs are worthwhile or if the current one is fully optimal.

Round 2

Reviewer 2 Report

Comments and Suggestions for Authors

The authors followed the comments.